# Investigation on Electrospun and Solvent-Casted PCL-PLGA Blends Scaffolds Embedded with Induced Pluripotent Stem Cells for Tissue Engineering

**DOI:** 10.3390/pharmaceutics15122736

**Published:** 2023-12-06

**Authors:** Mariella Rosalia, Martina Giacomini, Erika Maria Tottoli, Rossella Dorati, Giovanna Bruni, Ida Genta, Enrica Chiesa, Silvia Pisani, Maurilio Sampaolesi, Bice Conti

**Affiliations:** 1Department of Drug Sciences, University of Pavia, Viale Taramelli 12, 27100 Pavia, Italy; mariella.rosalia01@universitadipavia.it (M.R.); martina.giacomini01@universitadipavia.it (M.G.); erika.tottoli01@universitadipavia.it (E.M.T.); rossella.dorati@unipv.it (R.D.); ida.genta@unipv.it (I.G.); enrica.chiesa@unipv.it (E.C.); silvia.pisani@unipv.it (S.P.); 2Consorzio Interuniversitario per lo Sviluppo dei Sistemi a Grande Interfase (C.S.G.I.), Department of Chemistry, Physical Chemistry Section, University of Pavia, Via Taramelli 10, 27100 Pavia, Italy; giovanna.bruni@unipv.it; 3Translational Cardiomyology Laboratory, Head Unit of Stem Cell and Developmental Biology (SCDB), Head Department of Development and Regeneration, KU Leuven, ON4 Herestraat 49, Box 804, 3000 Leuven, Belgium; maurilio.sampaolesi@kuleuven.be

**Keywords:** electrospinning, solvent casting/particulate leaching, iPSC, polymer blends

## Abstract

The design, production, and characterisation of tissue-engineered scaffolds made of polylactic-co-glycolic acid (PLGA), polycaprolactone (PCL) and their blends obtained through electrospinning (ES) or solvent casting/particulate leaching (SC) manufacturing techniques are presented here. The polymer blend composition was chosen to always obtain a prevalence of one of the two polymers, in order to investigate the contribution of the less concentrated polymer on the scaffolds’ properties. Physical–chemical characterization of ES scaffolds demonstrated that tailoring of fibre diameter and Young modulus (YM) was possible by controlling PCL concentration in PLGA-based blends, increasing the fibre diameter from 0.6 to 1.0 µm and reducing the YM from about 22 to 9 MPa. SC scaffolds showed a “bubble-like” topography, caused by the porogen spherical particles, which is responsible for decreasing the contact angles from about 110° in ES scaffolds to about 74° in SC specimens. Nevertheless, due to phase separation within the blend, solvent-casted samples displayed less reproducible properties. Furthermore, ES samples were characterised by 10-fold higher water uptake than SC scaffolds. The scaffolds suitability as iPSCs culturing support was evaluated using XTT assay, and pluripotency and integrin gene expression were investigated using RT-PCR and RT-qPCR. Thanks to their higher wettability and appropriate YM, SC scaffolds seemed to be superior in ensuring high cell viability over 5 days, whereas the ability to maintain iPSCs pluripotency status was found to be similar for ES and SC scaffolds.

## 1. Introduction

Regenerative medicine aims to restore the physiologic functions of damaged and diseased tissues and organs through repair or replacement using therapeutic or stem cells, or by means of engineered issues and artificial organs. Tissue engineering focuses on the application of biomaterials for the design of scaffolds, intended as support for the expansion of cells and the de novo creation of endogenous tissue. The three cornerstones of tissue engineering are biomaterials, cells, and active molecules, which need to be wisely combined [1,2]. The definition of ideal biomaterial has changed over the last 50 years, moving from being “inert” to having “bio-inductive” features and integrating with adjacent tissues. Most of materials used to produce scaffolds for tissue engineering (TE) are polymers that must satisfy strict requirements, including good biocompatibility, mechanical properties, and processability. Biocompatibility depends on the biologic environment and tissue reaction after the scaffold is implanted in vivo [3,4]. Degradable biomaterials can be obtained by breaking them down into smaller molecules that can be reabsorbed by the human body, thus reducing the presence of foreign bodies in the organism and reducing their immunogenicity [2,5]. Biomaterials for TE can be classified based on their origin either synthetic or natural. Natural biomaterials show promising results in terms of biocompatibility and biodegradation, but have drawbacks such as scarce mechanical properties, high variability, and unavailability in large amounts. Synthetic biopolymers used in TE are biodegradable polyesters, including poly-lactic acid (PLA), poly-caprolactone (PCL), and copolyesters such as poly-lactic-co-glycolic acid (PLGA) and poly-lactic acid-co-caprolactone (PLA-PCL), which can be used as such or blended with both synthetic and natural materials [2,5].

Scaffold properties play a major role in tissue engineering (TE) product design and are closely associated with biomaterial features. The scaffold structure and architecture define its macroscopic porosity, ensuring nutrient delivery, cell-adhesion cues, and environmental protection [6]. The scaffold surface is crucial as it is the first interface between the scaffold, environment, and cells. A suitable TE scaffold can be obtained by carefully combining materials, scaffold manufacturing technique, and final scaffold architecture [7]. Electrospinning and solvent casting are popular techniques used to manufacture biodegradable scaffolds based on polyesters [8]. Electrospinning produces fibrous scaffolds with properties similar to extracellular matrix, while solvent casting/particulate leaching allows for the preparation of highly porous polymeric scaffolds. The particles are removed by submerging the film in water, leaving only the insoluble porous polymeric scaffold [9,10]. The main advantage of using the electrospinning technique is the fast and one-step and scalable preparation of scaffolds with an ECM-like structure, promoting cell–scaffold interaction and hydration of the scaffold, fundamental feature to ensure the exchange of wastes, nutrients and growth factors. On the other hand, solvent casting/particulate leaching is a less reproducible manufacturing technique due to phase separation within the polymer solution blend during the slow solvent evaporation steps and the inhomogeneous dispersion of the particulate, making scalability challenging. Nevertheless, very high pore interconnectivity can be achieved.

As for the cell source, Induced Pluripotent Stem Cells (iPSCs) are a powerful resource due to their similarity to Embryonic Stem Cells and their ability to differentiate towards various cell lines and tissue types. Over the past 17 years, iPSCs have been extensively studied for medical purposes, including disease modelling, drug screening, and regenerative medicine [11,12,13,14,15,16,17,18,19]. Several studies, some of which have advanced to clinical trials, have shown that iPSCs have the potential to generate functional tissues [19,20,21]. Several works on cardiac application (refs. [19,20,21]) can be found in the literature: Kaiser et al. used a fibrin and collagen scaffold to differentiate hiPSC-derived cardiac myocardium into engineered myocardium. The expression of cardiac troponin T (cTnT) in CM populations was found to be dependent on scaffold compaction. The findings provided a basis for creating specialised scaffold structures for cardiac tissue engineering by clarifying the relationship between scaffold interactions and hiPSC-derived CMs. [22]. Joanne et al. seeded iPSCs-derived cardiomyocytes on an electrospun collagen scaffold that was implanted in mice with non-ischemic dilated cardiomyopathy. Fourteen days after implantation, a stable heart function was reported in the treated mice compared with the controls. Moreover, they were able to increase the number of endothelial cells and promote scaffold vascularization [23]. Weinberger et al. created engineered human heart tissue (hEHT) strips from these cells, which were transplanted onto large defects in guinea pig hearts. After 28 days, the hearts showed human heart muscle grafts within the scar, showing cardiomyocyte proliferation, vascularization, and electrical coupling to the intact heart tissue. According to the study, hEHT strips resulted in a 31% improvement in left ventricular function when compared to pre-implantation, proving that the damaged heart can be repaired using three-dimensional human heart muscle constructs [24]. Nevertheless, the main challenge in using iPSCs is the potential occurrence of genetic alterations can lead to the formation of post-implantation teratoma. Pluripotency factors, such as NANOG, OCT4, and MYC, commonly used for pluripotency induction, have been associated with cancer progression [25,26]. Additionally, allogenic iPSCs therapy requires immunosuppression to avoid immune rejection, which can lead to higher infection risk [27]. Despite these challenges, there is an increase in iPSCs applications, and further investigation is necessary for them to become a life-changing tool. 

This work aims to design, produce, and characterise 3D scaffolds for cardiac tissue-engineering applications made of polylactic-co-glycolic acid (PLGA), polycaprolactone (PCL) and their blends, manufactured using electrospinning or solvent casting techniques, and to preliminarily investigate their suitability as iPSCs culturing support. The versatility of PCL and PLGA was already confirmed in tissue engineering application and their blending represents a valid strategy for the tailoring of polymeric scaffold properties. Specifically, this preliminary study has the purpose to characterise these scaffolds from a physical, chemical, and biological point of view and to correlate their composition and characteristics to their biological properties. A structural characterisation of the scaffolds through Scanning Electron Microscope (SEM) imaging, tensile tests, contact angle measurement and hydration assays were performed. Regarding the biological characterisation, cell seeding of iPSCs on selected scaffolds was performed and cellular viability was evaluated using XTT assay and correlated to the chemical physical scaffold properties and composition. Moreover, cell pluripotency and integrin expression were evaluated using Reverse Transcription PCR (RT-PCR) and Real Time quantitative PCR (RT-qPCR). The aim was to investigate if and/or how the scaffold architecture affects iPSCs viability and gene expression. Eventually, the results and discussion put a step forward in TE, and they provide useful information to the scientific community.

## 2. Materials and Methods

### 2.1. Materials

Polycaprolactone polymer (PCL, Mn 80,000) was purchased from Sigma-Aldrich-Merck (Milano, Italy) and polylactic-co-glycolic acid (PLGA, Resomer LG 824 82:18 lactic acid:glycolic acid) was supplied by Evonik Industries (Essen, Germany). Dichloromethane (DCM), *N*,*N*-Dimethylformamide (DMF), 1,4-dioxane, Sodium Chloride (NaCl), Potassium Chloride (KCl), Magnesium Sulphate (MgSO_4_), Monosodium Phosphate (NaH_2_PO_4_), saccharose and D-glucose anhydrous were purchased from Carlo Erba Reagents (Cornaredo, Milan, Italy), and spray dried lactose from Meggle Pharma-Excipients & Technology (Wasserburg am Inn, Germany). The dialytic membrane Spectra/Por 2 (molecular weight cut-off: 12–14 kDa) was supplied by Sigma-Aldrich-Merck (Milano, Italy); Geltrex LDEV-Free hESC-qualified Reduced Growth Factor Basement Membrane Matrix, Essential 8 Flex cell culture medium kit, DMEM/F12 medium, Knockout serum, L-Glutamine, HEPES, Penicillin-Streptomycin (10,000 U/mL) and TrypLE Express enzyme were purchased from Thermo Fisher Scientific (Rodano, Milan, Italy). Induced Pluripotent Stem Cells (iPSCs) were a commercially available Episomal hiPSC line from Gibco (Thermo Fisher Scientific, Rodano, Milan, Italy), generated from cord blood derived CD34+ cells, using three plasmids carrying seven episomally expressed factors, namely OCT4, SOX2, KLF4, MYC, NANOG, LIN28 and SV40T.

### 2.2. Methods

#### 2.2.1. Scaffolds Preparation by Electrospinning

The scaffolds were prepared with a NANON 01A Electrospinner (MECC Co., Instruments, Fukoka, Japan) equipped with a dehumidifier and a plane collector. The polymer solution was extruded from a 5 mL syringe with a 25-gauge needle; the needle-to-collector distance was 15 cm; the humidity was 23% RU; and the temperature was kept between 30 and 35 °C. The polymer solutions to be electrospun were PCL 14% *w*/*v* in DCM/DMF 75:25, PLGA 10% *w*/*v* in DCM/DMF 75:25, and their combinations, as shown in Table 1. When PCL was the prevalent polymer, i.e., batches ES1, 2, and 3, the final polymer blend concentration was 14% *w*/*v*, while PLGA was the prevalent polymer, i.e., batches ES4, 5, and 6, the final polymer blend concentration was 10% *w*/*v*. The blend compositions were designed to keep the main polymer concentration as close to the critical entanglement concentration of the specific polymer as possible. Preliminary trials were carried out on single polymer solutions, testing a variety of process conditions (Appendix A). The final electrospinning conditions used for the production of electrospun matrixes were 15 kV voltage and 0.1 mL/h feed rate for an electrospinning time of 30 min. The residual solvent was removed with evaporation under chemical hood for 16 h. The scaffolds were stored at −25 °C for at least 2 h before gently detaching the electrospun matrix from the aluminium foil and stored at room temperature for further characterisation.

#### 2.2.2. Scaffolds Preparation by Solvent Casting/Porogen Leaching

Solvent casting with the porogen leaching method involves the manufacturing of polymer films by casting polymer solutions into moulds and evaporating the solvent. The moulds contain solid particles of defined sizes acting as porogen, that should be eventually eliminated through dissolution, obtaining a porous polymeric scaffold. Several statistical representative trials were carried out using NaCl, glucose and spray-dried lactose as porogen materials (Appendix A) and spray-dried lactose was the finally selected porogen. The scaffolds were prepared starting from the same polymer blend used for electrospinning as reported in Table 1. Briefly, polymer pellets at a concentration of 10% *w*/*v* were swollen in 100% dioxane until complete dissolution. Then, 35% *w*/*v* spray-dried lactose was added to the polymer solution and mixed thoroughly until a uniform suspension was obtained. The polymer–porogen mixture was poured and uniformly distributed in a 10.5 cm diameter Teflon mould and the solvent was evaporated under a chemical hood for 16 h. The dry polymer film was detached from the mould and the porogen was removed via dialysis using a Spectra/Por 2 dialytic membrane (MWCO: 12–14 kDa) soaked in 100 mL of purified water. The water bath was completely renewed in 6 h, and dialysis was continued up to 16 h. The dialyzed scaffolds were freeze dried with a LIO 5P freeze dryer (Edwards, Milan, Italy) at −50 °C and a pressure of 0.4 mbar for 16 h. The dry scaffolds were stored at room temperature for further characterisation.

#### 2.2.3. Scaffolds’ Physicochemical Characterisation

##### Scanning Electron Microscopy (SEM)

To evaluate the nanometric morphology of the scaffolds, a 2D-image-based method was applied. SEM was performed on both electrospun and solvent-casted samples, fixed with carbon tape on a stud, and gold sputtered. SEM analyses were performed with a ZEISS EVO MA 10 SEM microscope (ZEISS Group, Milan, Italy). Images were processed with ImageJ software version 1.54f (U.S. National Institutes of Health, Bethesda, MD, USA) for the determination of electrospun fibres diameter, scaffolds’ surface pores diameter, and apparent percentage porosity. The line tool was used to measure fibre diameter. NiBlack thresholding was used to obtain local threshold images for overall percentage porosity analysis. Particle Analyser tool was applied to measure the pores’ diameter.

##### Mechanical Uniaxial Tensile Testing

For the mechanical tensile testing, both electrospun and solvent-casted scaffolds were cut into a standard dog bone shape following ASTM D638 guidance for the mechanical tensile testing of plastics using a manual punching machine (Noselab Ats, Bovisio Masciago, Monza-Brianza, Italy). The dog-bone specimens were secured between the jaws of a MARK-10 ESM303 force tester (Wagner Instruments, Greenwich, UK) and stretched until rupture at 612.1 mm/min, based on the beating rate of human heart (70 bpm). The resulting Load (N) vs. Travel (mm) curves were recorded and transformed to Stress (Mpa) vs. Strain (mm/mm) curves using the following Equations (1) and (2):Stress (Mpa) = F/A(1)
Strain (mm/mm) = D/L_0_(2)
where F is the load (N) exerted by the loading cell; A is the sectional area of the specimen (mm^2^), calculated by multiplying the width (mm) and thickness (mm) of the specimen; D is the displacement of the jaws (mm) from the zero position; and L_0_ is the initial length (mm) of the specimen. The length and width were 15 mm and 4 mm, based on the standardised cutting die, while thickness of the specimen was measured with a digital calliper. Stress vs. strain curves were used to calculate the Young’s modulus (YM, in MPa) as the slope of the initial linear portion of the stress vs. strain plot, which describes the elastic behaviour of the sample [28].

##### Contact Angle Measurements and Hydration Assays

Contact angle measurements were performed on the solvent-casted and electrospun samples using a Contact Angle Meter Dme-211 (Kyowa Interface Science, Saitama, Japan), equipped with a sample stage, a glass syringe, a light source and a camera. The 2 × 2 cm specimens were secured with tape on the instrument’s stage and a 10 μL bi-distilled water drop was gently deposited on the sample; FAMAS software version 3.3 (Kyowa Interface Science, Saitama, Japan) was used to acquire images of the water drop and calculate the contact angles. 

Hydration of electrospun and solvent-casted samples was evaluated in artificial interstitial fluid prepared, according to the method proposed by Bollella and co-workers [29], at pH = 7.4 and 37 °C for one week. To prepare the artificial interstitial fluid, 3.5 mM potassium chloride, 123 mM sodium chloride, 1.5 mM sodium dihydrogen phosphate, and 7.4 mM saccharose were dissolved in distilled water and the pH was adjusted to 7.4. The 2 cm × 2 cm dry scaffolds were weighed, secured with Cell Crowns (Scaffdex Oy, Tampere, Finland) on the bottom of a 6-wells plate and submerged with 8 mL of artificial interstitial fluid. The samples were incubated at 37 °C and at each time point (1, 3, 5, 6.5, 16, 24, 48, 72, 144, and 168 h) the scaffolds were blotted on tissue paper and weighed. The average normalised water uptake (WU) was calculated using Equation (3):WU (%) = (m(t) − mi)/mi × 100(3)
where m(t) is the mass of the specimen at the sampling point and mi is the initial mass of the specimen. 

#### 2.2.4. Scaffolds’ Preliminary Biological Characterisation

##### Cell Culture on Electrospun and Solvent Casted/Particulate Leaching Scaffolds

Episomal hiPS cells were cultured on both electrospun and solvent-casted sample disks of 2 cm in diameter. The scaffolds were sanitised before use by soaking them for 3 min in 70% ethanol solution, then rinsed with sterile PBS and stored in PBS with 1% Penicillin-Streptomycin. Prior to use, the samples were coated with 800 uL of Geltrex coating solution. The hiPSCs were cultured in complete E8 Medium (following the supplier’s instruction for preparation of the media), at 37 °C in normoxia conditions. Cell culture medium was changed daily until 80% cell growth confluence was reached; cells were detached with TrypLE enzyme solution and seeded at a concentration of 2.5 × 10^4^ cells/cm^2^ on Geltrex-coated scaffolds placed in a 12-well plate. The seeded samples were incubated at 37 °C in normoxia conditions, and at different time points (1, 3 and 5 days), XTT assay, RNA extraction and, RT-qPCR analysis were performed.

##### Cell Viability Assay

The XTT (2,3-bis-(2-methoxy-4-nitro-5-sulfophenyl)-2H-tetrazolium-5-carboxanilide) analyses were performed in triplicate at 3 timepoints (1, 3, and 5 days after seeding) on both scaffolds and controls. The seeded scaffolds for XTT analysis were transferred in new 12-well plate and treated with activator and reagent solutions from the TACS XTT Cell Proliferation Assay kit (R&D systems, Fisherscientific, Waltham, MA, USA) according to manufacturer’s instructions. Absorbance was read at 450 nm and 630 nm with a Synergy HTX Plate Reader (BioTek, Agilent, Leuven, Belgium). The absorbance values for each time point were recorded, and the cell viability percentage was calculated with the following Equation (4), using hiPSCs cultured in 12-well plates as control.
Cell Viability (%) = (Sc Abs_450–630_ − Blk Abs_450–630_)/(Ctrl Abs_450–630_ − Blk Abs_450–630_) × 100(4)
where Sc Abs is the XTT absorbance of XXT-treated cells cultured on the scaffolds, blk Abs is the absorbance of the blank, and Ctrl Abs is the absorbance of XTT-treated control cells. 

##### RNA Extraction and Reverse Transcription-PCR

Episomal hiPS cells cultured on electrospun and solvent-casted/particulate leaching scaffolds were detached from the scaffolds by adding 500 µL TripLE Express (5 min at 37 °C), and the enzyme was neutralised with 1 mL of warm DMEM/F12 + L-Glutamine + HEPES supplemented with 20% knockout serum. Detached cells were collected in a tube and centrifuged at 1200 rpm for 4 min. The obtained pellet was resuspended in 300 µL of lysis buffer and 300 µL 70% ethanol. Then, 650 µL of lysate were collected and RNA was extracted using PureLink RNA Mini kit (Invitrogen, Thermo Fisher Scientific, Dilbeek, Belgium) according to manufacturer’s instructions. The collected RNA was purified from DNA using a TURBO DNA-Free kit (Invitrogen, Thermo Fisher Scientific, Dilbeek, Belgium) according to manufacturer’s instructions. The sample containing the extracted and purified RNA was set on ice, and the amount of RNA was measured with a NanoDrop 1000 spectrophotometer (Thermo Fisher Scientific, Rodano, Milan, Italy). This protocol was repeated for RNA extraction from each sample (n = 3), and the results are displayed as ng/µL. The extracted RNA underwent Reverse Transcription Polymerase Chain Reaction (RT-PCR) protocol in order to retrotranscribe the RNA into cDNA. The Biometra T3000 Thermocycler (Biometra GmBH, Göttingen, Germany) was used. Samples were incubated for 10 min at 25 °C, 30 min at 50 °C, and the reaction was terminated after 5 min at 85 °C.

##### Quantitative Real-Time Polymerase Chain Reaction (RT-qPCR)

RT-qPCR was performed to evaluate iPSC pluripotency and adhesion gene expression. The expression of 7 genes, namely GAPDH, NANOG, OCT4, ITGα6, ITGβ1, ITGβ5 and ITGαV, at three timepoints (1, 5 and 5 days) was evaluated. GAPDH was used as housekeeping gene and taken as reference gene for the relative expression (ΔCt) calculation. The Forward and Reverse primers of each gene used in the analysis are listed in Table 2. Sample preparation and RT-qPCR were performed as follows. Briefly, 5 µL of the RT-PCR cDNA samples were placed in each well of a 384-well plate, along with 5 µL of master mix (5 L of SYBR Green + ROX solution, and 1 µL Forward and Reverse primers solution from 100 µM to 2.5 µM in DEPC water). The 384-well plate was sealed with transparent film and centrifuged at 1000 rpm for 1 min before being incubated for 40 cycles at 95 °C for 15 s and 60 °C for 45 s in the ViiA7 Real-Time PCR system (Applied Biosystems, Thermo Fisher Scientific, Dilbeek, Belgium). The Ct values for each time point were recorded, and the Delta Ct values (ΔCt) calculated by subtracting the Ct values of the genes of interest to the Ct values of the housekeeping gene (GAPDH). Each sample was analysed twice.

#### 2.2.5. Statistical Analysis

All data were processed with statistical analysis using GraphPad Prism version 8 software (San Diego, CA, USA). The data obtained from ImageJ, mechanical tensile testing, contact angle, water uptake measurements and XTT assay were analysed using one-way ANOVA analysis with Tukey’s multiple comparison, whereas iPSC pluripotency and adhesion gene expression data were statistically analysed using the one-way ANOVA analysis with Sidak’s correction. The significance of the results was determined by *p*-value (ns = *p* > 0.05, * = *p* ≤ 0.05, ** = *p* ≤ 0.01, *** = *p* ≤ 0.001, **** = *p* ≤ 0.0001). All results were plotted as the average value ± standard deviation.

## 3. Results

### 3.1. Morphometric Characterisation of Electrospun and Solvent-Casted Scaffolds

As shown in Table 1, electrospun scaffolds were created by electrospinning various PCL and PLGA solution blends prepared in a DCM/DMF solvent system. The results of the morphometric analysis performed with SEM microscopy at 1000× and 3000× magnifications are reported in Figure 1. The results of ImageJ measurements are reported in Table 3, while fibre diameter distributions are reported in Appendix A.

SEM images showed an irregular surface for 100% PCL fibres (ES1), whereas 100% PLGA fibres (ES4) were stretched and smooth. Moreover, an increase in PLGA content in PCL-predominant electrospun matrixes (ES2 and ES3) led to the formation of a heterogeneous population of fibres with highly variable diameter. On the contrary, the addiction of PCL to PLGA predominant electrospun matrixes (ES5 and ES6) did not affect the fibre homogeneity while demonstrating an influence on fibre diameter. In fact, PCL-predominant fibres (ES1, ES2, and ES3) had an average diameter close to 1 μm, whereas PLGA-predominant fibres (ES4, ES5 and ES6) had submicrometric dimensions and a significant increase in fibre diameter was reported with the increase in the PCL content, especially in ES6 samples (Figure 2). It is also interesting to underline that the PLGA 100% scaffold (ES4) displayed the thinnest fibres among all the scaffolds, whereas the biggest fibres were obtained with the PCL 85% + PLGA 15% polymer blend (ES3). Due to the high-density fibres deposition, small pores around 3 μm in diameter were obtained for all samples and no significant variation in the average pore diameter was reported. Concerning overall porosity, higher percentages were reported for PLGA-predominant electrospun matrixes (ES4, ES5 and ES6), probably related to the smaller fibre diameters.

Scale-up and production of scaffolds with the solvent casting technique were performed using dioxane polymer solutions of PCL and PLGA blends, whose composition is reported in Table 1. The results of the scaffolds surface morphometric analysis performed through SEM microscopy at 2000× and 3000× magnifications are reported in Figure 3. The results of ImageJ measurements are reported in Table 3. SEM images of the scaffolds surface clearly show a “bubble-like” topography due to the spherical shape of spray-dried lactose that was used as porogen. Images at 3000× magnification showed high variability in terms of microporosity by varying the polymer composition in the casting solution. Scaffolds predominantly made of PCL (SC1, SC2, and SC3) displayed a more uniform pore distribution, higher average pore diameter and porosity percentage. Increasing PLGA concentration in PCL-predominant scaffold led to an increase in pore diameter; hence, SC3 (PCL 85% + PLGA 15%) displayed the highest average pore diameter, whereas SC1 (100% PCL) showed the highest overall porosity. The worst results were obtained for scaffolds predominantly made of PLGA (SC4, SC5 and SC6) that showed low or absent surface porosity and smaller pores. Especially scaffold SC4, made of 100% PLGA, was excluded from further characterisation because scatter and almost absent porosity was highlighted via SEM. Overall, higher PLGA content in solvent-casted films resulted in lower porosity percentage and more heterogeneous pore distribution. These results confirmed the peculiarities of each manufacturing technique in achieving scaffolds with different characteristics in terms of porosity percentage, average pore diameter, and homogeneity starting from the same polymer solutions.

### 3.2. Mechanical Properties of Electrospun and Solvent-Casted Scaffolds

In both electrospun and solvent-casted samples, the mechanical properties were mainly influenced by the predominant polymer of the blend, as displayed by YM reported in Table 4: higher YM were reported for scaffolds with higher PLGA content, as it is a polymer with stiffer and more brittle behaviour, whereas scaffolds with predominant PCL content resulted in lower YM, according to the polymer’s softer and more ductile nature. The statistical analysis (Figure 4a) displayed a non-significant difference between the YM of electrospun scaffolds whose composition is mainly PCL (batches ES1, ES2, ES3), meaning that PLGA addition up to 15% *w*/*w* did not affect the scaffold YM. Interestingly, the analysis also showed a significant difference between batch ES4 (PLGA 100%) and batches ES5 and ES6 (PLGA 85% + PCL 15% and PLGA 95% + PCL 5%, respectively). This result is consistent with PLGA’s higher stiffness compared to PCL and suggests that blending PCL with PLGA in amounts greater than 5% significantly changes scaffold mechanical properties. Consistent with the highly significant difference between the YM of scaffolds ES1 and ES4 (PCL 100% and PLGA 100%), PCL addition to PLGA leads to drastically lowering YM values, which confirms the plasticizing effect of PCL. Solvent-casted scaffolds’ mechanical behaviour was not comparable to that of electrospun scaffolds (Table 4), as expected because of the different morphologies. The results of statistical analysis performed on solvent-casted scaffolds did not show any significant difference between their YM (Figure 4b), with the exception of SC2 and SC5, suggesting that changes in polymer composition do not correlate with a change in YM value when a solvent casting manufacturing technique is used. The manufacturing technique led to inhomogeneous polymer composition of the solvent-casted scaffolds resulting in scattering tensile properties that made the correlation of YM with the scaffolds’ polymer composition challenging.

### 3.3. Surface Wettability and Water Uptake of Electrospun and Solvent-Casted Scaffolds

The values of contact angles of all electrospun scaffolds were above 90° (Table 4), showing hydrophobic properties. Statistical comparison shown in Figure 5 demonstrated that a PLGA content higher than 15% decreased the wettability of electrospun scaffolds with a predominant PCL content. A similar trend was reported when the PCL content of PLGA-based electrospun scaffolds reached 15%. On the other hand, opposite behaviour was reported for solvent-casted scaffolds, whose wettability improved with the increase in the PLGA content above 15%, as proven by contact angle values below 90° for SC3 (PCL 85% + PLGA 15%), SC5 (PLGA 95% + PCL 5%), and SC6 (PLGA 85% + PCL 15%). Overall, solvent-casted scaffolds had greater wettability in comparison to electrospun scaffolds. As far as hydration is concerned, it appears evident from the two plots in Figure 5 that electrospun scaffolds exhibit a much higher water uptake capacity compared to the solvent-casted scaffolds. The PLGA 100% ES4 scaffold uptakes the highest amount of artificial interstitial fluid, followed by ES3 (PCL 85% + PLGA 15%), ES5 (PLGA 95% + PCL 5%) and ES6 (PLGA 85% + PCL 15%) that showed comparable hydration profiles; the lowest water uptake was reported for PCL-based scaffolds ES1 (PCL 100%) and ES2 (PCL 95% + PLGA 5%). The hydration profiles of solvent-casted scaffolds are ten-fold lower than that of electrospun ones and less scattering between solvent-casted samples hydration profiles was reported, suggesting superimposable water uptake properties. Among all samples, 100% PCL SC1 scaffold displayed the lowest hydration rate.

### 3.4. Rationale of Scaffold Selection for Preliminary Biological Characterisation

To significantly assess the biological performance of electrospun polymeric scaffolds, batches showing opposite physicochemical properties but suitable for potential use in tissue regeneration were tested. Hence, ES1 (PCL 100%) and ES3 (PCL 85% + PLGA 15%) electrospun scaffolds were selected considering that the two scaffold types had comparable morphology and mechanical properties suitable for soft tissue engineering applications. On the other hand, ES1 has a lower contact angle and poorer water uptake compared to ES3, which displays higher hydrophobicity but also higher hydration capacity. In the case of solvent-casted samples, since no significant physicochemical differences were observed by varying the polymer composition, selection of solvent-casted samples for the biological tests was performed on the basis of YM values. Hence, batches SC3 (PCL 85% + PLGA 15%) and SC5 (PCL 95% + PLGA5%) were selected, which have double YM between each other. Moreover, in addition to the different architecture between solvent-casted and electrospun scaffolds, SC3 and SC5 displayed hydrophilic properties, in contrast to the selected electrospun samples. Following this selection strategy, a high range of morphological features, mechanical properties, wettability, and hydration features were covered.

### 3.5. iPS Cell Viability on Electrospun and Solvent-Casted Scaffolds

The results of XTT test were elaborated using statistical analysis and are plotted in Figure 6, where graphs in Figure 6a,b highlight different statistical comparisons. Overall, higher cell viability was obtained for samples ES1 (100% PCL) and SC3 (PCL 85% + PLGA 15%), and especially the solvent-casted SC3 scaffold was the only one ensuring cell viability around 70% for all three time points. The result highlighted that the used polymer blends did not thrive on iPSC growth, and not even Geltrex coating led to an improvement; however, a trend towards PCL being a better substrate for iPSCs was reported. This assumption is strengthened by the statistical analysis showed in Figure 6a, in which scaffolds manufactured with the same techniques, but with different polymer compositions are compared. Significant decrease in cell viability between SC3 and SC5 (days 1 and 5) and also between ES1 and ES3 (days 3 and 5) was shown and could be related to an increase in the PLGA content of the scaffold. The results of statistical comparison among electrospun and solvent-casted scaffolds over the three time points are shown in Figure 6b. Over time, solvent-casted scaffolds performed overall better than electrospun samples, and this was especially true for later time points and for SC3 (PLGA 85% + PCL 5% polymer blend).

### 3.6. iPSC Pluripotency and Adhesion Genes Expression

The RT-qPCR analysis results comparing the expression of the genes by the cells seeded on the ES1 (PCL 100%) scaffold at all three timepoints are reported in Figure 7. Overall, no significant difference in the expression in any of the tested genes was shown, meaning that the gene expression is maintained stable throughout the 5 days of incubation of iPSCs with ES1 as reported in Figure 7d. Going into details of each investigated timepoint, no significant differences in gene expression between the control and the scaffolds are shown on day 1. Therefore, it can be assumed that gene expression of all six genes analysed between the scaffold and the control are comparable (Figure 7a). The pluripotency status of iPSCs both in the controls and ES1 scaffold was kept up to the third day, with a slightly significant difference in the ITGβ1 expression (Figure 7b). While ITGβ1 expression in the control cells has decreased since day 1, it has increased in the scaffold-seeded cells. The most relevant result is the higher expression of the β1 subunit from the ES1 scaffold-seeded iPSCs compared to the control on the fifth day of incubation (Figure 7c), which increased significantly with respect to day 3, suggesting an overall high expression of ITGβ1.

The comparison of RT-qPCR results of cells seeded on SC3 (PCL 85% + PLGA 15%) scaffolds at the three evaluated timepoints is reported in Figure 8d and showed no significant difference in gene expression of the six genes throughout the 5 days, the same as for the ES1 scaffold. The detailed comparison on day 1 between the pattern of gene expression of iPSCs incubated on SC3 and control was very similar to that of ES1 with the cells in a condition of pluripotency: ITGα6, αV, and β5 displayed very similar gene expression values, and ITGβ1 showed higher expression, even if it was not significantly different (Figure 8a). On day 3, significant differences in the expression of NANOG were highlighted, resulting in it being more expressed in the SC3 scaffold seeded cells rather than the control (Figure 8b), while integrins and OCT4 expression kept stable without any significant difference between the scaffold and control cells. Eventually, on day 5, the results of the gene expression comparison vs. control (Figure 8c) showed no significant difference in the expression of OCT4 and integrins, with only a slight difference in NANOG expression. However, it should be highlighted that there was no significant difference in NANOG expression in SC3 scaffold-seeded cells throughout the 5 days of analysis. It should also be taken into consideration that the pattern of expression on day 5 is very similar to the one displayed at the same timepoint by the ES1 scaffolds.

## 4. Discussion

Both electrospinning and solvent casting are consolidated techniques to manufacture biodegradable polymer scaffolds based on polyesters. Both techniques have pros and cons for their application in cardiac tissue regeneration: fibrous scaffolds are potentially able to induce anisotropic orientation of attached cells, while highly porous (50–90%) scaffolds obtained using solvent casting/particulate leaching technique ensure cell migration, nutrients, and oxygen exchange [30]. In previous works, we thoroughly evaluated both electrospinning and solvent casting/porogen leaching process parameters to work out polylactide and polylactide-co-polycaprolactone polymers [31,32,33,34,35,36]. Our previous investigations led us to focus on PLGA and further investigate its blending with polycaprolactone in variable ratios, as shown in Table 1. The aim of blending PLGA and PCL was to exploit their different chemical and mechanical properties, which, when combined with appropriate manufacturing techniques could lead to the production of scaffolds with adequate properties for cardiac tissue engineering. In fact, PLGA is characterised by good mechanical strength and is capable of resisting intense mechanical stresses, such as in muscular contraction; on the other hand, PCL is a rubbery biomaterial with elastomeric-like features and is characterised by a good compliance toward soft tissue. By combining the two materials, adequate mechanical strength and compliance could be obtained, even in thin scaffolds (<100 µm) [37]. The polymer blend composition was chosen to always have a prevalence of one of the two polymers in order to investigate the contribution of the less concentrated polymer and better highlight its influence on the scaffolds’ properties. For the preparation of electrospun matrixes, a DCM/DMF 75:25 solvent system was selected to ensure both optimal polymer dissolution (DCM) and enhanced polymer solution conductivity (DMF) [32]. Different overall polymer concentrations were used depending on the prevalent polymer type, namely 14% and 10% *w*/*v* for PCL and PLGA. In fact, the selected polymers had different molecular weights, and thus resulted in them not having superimposable viscosities in the selected solvent system when the same concentration was used. Viscosity studies (Appendix A) permitted the identification of the critical entanglement concentration (CEC) [38] of the two polymers in DCM/DMF 75:25 solvent system, 7.47% *w*/*v* for PCL and 6.70% *w*/*v* for PLGA. Several electrospinning trials with polymer solutions having concentrations above the CEC were performed to determine the optimal polymer concentration and electrospinning parameters. Statistically analysed fibre morphology data did not display any significant difference between the fibre diameters of the PCL-based scaffolds ES1, ES2, and ES3 (PCL 100%, PCL 95% + PLGA5%, and PCL 85% + PLGA 15%, respectively), meaning that the addition of 5% and 15% PLGA to the polymer solution to be electrospun did not affect the fibre diameters. On the other hand, irregular fibre formation was observed for ES2 and ES3. Nonetheless, the statistical analysis showed more interesting results concerning the PLGA-based scaffolds. In particular, a significant difference was highlighted in the diameters between ES5 (PLGA 95% + PCL 5%) and ES6 (PLGA 85% + PCL 15%) scaffolds, meaning that the increase in the fibre diameters is related to the increased PCL concentration. This assumption is corroborated by the highly significant difference in fibre diameters between ES4 (PLGA 100%) and ES6 (PLGA 85% + PCL 15%) scaffolds (see Table 3 and Figure 2), underlying how the addition of PCL to PLGA increased the electrospun scaffold fibres diameters. It is well known that the electrospinning process is influenced by several parameters, including solution properties and composition, applied voltage, type of current, polymer solution flow rate, needle internal diameter and distance from collector, and type of collector [8,39,40]. Among solution parameters, polymer concentration and the resulting viscosity are considered the main factors in determining fibres diameter [41]: PCL 100% *w*/*v* solution in DCM/DMF 75:25 had a zero-shear viscosity of 1.023 ± 0.030 Pa×s, whereas PLGA 100% *w*/*v* solution in the same solvent system had a viscosity of 0.454 ± 0.019 Pa×s (Appendix A), explaining the smaller fibres obtained from PLGA solutions. Moreover, as already reported for the PLA/PCL blends [42], the polymers might not be perfectly thermodynamically miscible, and this influences fibres formation and their diameter. In fact, if polymer phase separation occurs during electrospinning, due to immiscibility and increased interfacial energy, irregular fibre diameter as reported for ES2 and ES3 (PLGA 95% + PCL 5% and PLGA 85% + PCL 15%) or increased fibre diameter as reported for ES5 and ES6 (PCL 95% + PLGA 5% and PCL 85% + PLGA 15%) can occur. Regarding porosity, no significant changes in pore size were reported between the different scaffolds; the high standard deviation is related to high fibre deposition density that led to the formation of both micro- and microporosity and therefore to heterogeneous pore size. For percentage porosity, no significant differences were reported, even if a trend towards higher porosity percentage by lowering fibre diameter was observable. Further confirmation that electrospinning process is influenced by several parameters can be drawn through a comparison with the study by Bazgir and co-workers [43]. They investigated electrospun scaffolds made from PCL and PLGA but under different process conditions and polymer parameters than the ones reported in this work. As a consequence, the characteristics of the electrospun matrices manufactured by Bazgir and coworkers are significantly different to those of the electrospun matrices characterized here, demonstrating how the process conditions and polymer solution compositions are fundamental to the obtained result. 

Solvent-casted samples were prepared by dissolving 10% *w*/*v* polymer or polymer blend into dioxane. Dioxane was preferred to halogenated solvents because it has a higher boiling point (101 °C), therefore limiting evaporation during manipulation and casting procedures. The solvent is miscible with water and has a high freezing point (12 °C), which allows for the removal of residual dioxane during both porogen leaching step in deionised water and lyophilization procedure. The same polymer blends as in electrospinning scaffolds were used (Table 1). The scaffold surface had a “bubble-like” structure caused by the polymer deposition on the surface of the spherical spray-dried lactose particles, promoting the formation of highly interconnected pores. PLGA-rich scaffolds, i.e., SC3 (PCL 85% + PLGA 15%) and especially SC4 (PLGA 100%), were more likely to form a continuous layer on the scaffolds’ surface, negatively influencing the porosity percentage. In the case of SC4, the scattering and almost absent porosity was the reason for discarding this type of scaffold from any other characterisation. The reduction in surface porosity could be attributed to the different matter phase of the polymers in the working conditions: solvent casting is performed at room temperature (20 °C), whereas lyophilization is performed at −50 °C, and since the glass transition temperature and the melting temperature of PCL are −60 °C and 60 °C, respectively [44], PCL was in its rubbery/viscous state during both moulding and freeze drying procedures. Therefore, it was more likely to accommodate the lactose particles’ shape, leading to the “bubble-like” structure. In the case of PLGA, whose glass transition temperature and melting temperature are between 54–60 °C and 157–164 °C, respectively, as reported by the manufacturer (Product Technical sheet; https://healthcare.evonik.com/en), the polymer was in its glassy state and therefore formed a smoother surface. Moreover, the microporosity reported for PCL-richer samples, i.e., SC1 (PCL 100%), SC2 (PCL 95% + PLGA 5%), SC5 (PLGA 95% + PCL 5%), and SC6 (PLGA 85% + PCL 15%), might be the result of mechanical stress on the polymer film caused by the formation of small ice crystals inside the wet dialysed scaffold. During sublimation, pore destabilisation and rearrangement of rubbery PCL could occur, thereby resulting in microporosity. Conversely, the stiff structure of PLGA hinders this phenomenon and impedes or lowers the formation of micropores [31].

Summing up, polymer scaffolds can be obtained through both techniques, but electrospinning yields more reproducible results, especially in terms of morphology regularity, as shown in Table 3. Depending on the type of polymer, significantly different morphologies can be achieved within same manufacturing technique; this is especially true for the solvent casting technique, as shown in Figure 3. The use of polymer blends can be a useful strategy to adapt the manufacturing technique to the biomaterial properties and obtain suitable tissue engineering scaffolds. The influence of polymer raw material properties on the final scaffold morphology is a fundamental aspect to take into consideration when choosing a biomaterial. The biocompatibility and the biological and mechanical performance of the scaffolds are the ultimate results of the interplay between both polymer and scaffold architecture.

The scaffolds’ physicochemical properties, i.e., Young modulus, contact angle, and hydration profile, are significantly affected both by scaffold composition and manufacturing technique. As far as tensile strength is concerned, the results showed that electrospun scaffolds were in keeping with the mechanical properties of PCL and PLGA, and Young moduli values reported in Table 4 are in accordance with the literature [45,46]. PCL is a rubbery-like material with low mechanical strength that acts as a plasticizer lowering YM, whereas PLGA is a rigid polymer with high mechanical strength that leads to higher YM in PLGA-predominant electrospun scaffolds. Moreover, a relation between mechanical strength and fibre diameter was already reported in literature: by increasing fibre diameter, a decrease in the elastic modulus has to be expected [47,48]. Indeed, a decrease in YM was accompanied by an increase in fibre diameter in PLGA-predominant electrospun scaffolds (see Figure 2 and Figure 4). Hence, both the polymer properties and the fibres morphologies were determinant for the mechanical performance of the electrospun scaffolds. In the case of solvent-casted samples, variable results were obtained, demonstrating that the solvent casting process led to an inhomogeneous scaffold composition. PCL-predominant scaffolds had smaller YM moduli compared to PLGA-based ones, but a relationship between polymer composition and mechanical strength was not reported for SC1 (PCL 100%), SC2 (PCL 95% + PLGA 5%), and SC3 (PCL 85% + PLGA 15%). For PLGA predominant scaffolds, i.e., SC5 (PLGA 95% + PCL 5%) and SC6 (PLGA 85% + PCL 15%), a reduction in YM with the increase in PCL content was observed and was in keeping with PCL plasticising features. The inhomogeneous composition of the scaffolds could be caused by phase separation of the blends during slow solvent evaporation, leading to the not-uniform deposition of polymer chains. Depending on the intended use, mechanical properties of tissue engineering scaffolds need to match the native tissue characteristics. The YM values shown in Table 4 are in keeping with those of cardiovascular tissues, including heart tissues such as the pulmonary valve (16.05 ± 2.02 MPa) and the aortic valve (15.34 ± 3.84 MPa), blood vessels, human femoral artery (circular YM 9–12 MPa), internal mammary artery (circular YM 8 MPa, longitudinal YM 16.8 MPa), and saphenous vein (circular YM 4.2 MPa, longitudinal YM 23.7 MPa) [49], but also other soft tissues such as ligaments related to the uterus, including the cardinal (0.5–5.4 MPa), the round (9.1–14.0 MPa) and the uterosacral ligament (0.75–29.8 MPa), cartilage (15 MPa) [50], tissues from the oral cavity including hard palate (18.13 ± 4.51 MPa) and buccal mucosa (8.33 ± 5.78 MPa) [51]. Especially PLGA-based electrospun samples and solvent-casted samples reported high YM compatible with soft tissue subjected to intense loads, such as cardiovascular or articular tissue.

The contact angle measurements can be interpreted in light of the fact that both PCL and PLGA are hydrophobic polymers, with PLGA being slightly more hydrophilic than PCL due to its free hydroxylic functional groups. In the electrospun scaffolds, an increase in PLGA content did not improve the samples’ wettability, and values were all above 90° showing hydrophobic behaviour [52]. On the contrary, an increase in contact angle values was reported by increasing the PLGA content of PCL-based scaffolds ES2 and ES3. A possible explanation is that the electrospun matrix’s wettability is influenced not only by polymer type but also by fibre distribution and their diameter. In the literature, as reported by Sadeghi P. and coworkers, an increase in contact angle values was related to greater fibre diameter [53]. If the average fibre diameters ± standard deviation of ES2 (1.085 ± 0.360 μm) and ES3 (1.076 ± 0.537 μm) samples were compared, a slight increase in fibre size for less wettable ES3 was reported. This trend was even more evident for ES5 (0.740 ± 0.181 μm) and ES6 (0.969 ± 0.195 μm), in which a significant increase in fibre diameter was accompanied by a significant increase in hydrophobicity. The opposite trend was highlighted for solvent-casted samples: an increase in PLGA content led to a gradual decrease in contact angle values, resulting in contact angle values lower than 90° and therefore hydrophilic properties [52]. The lower contact angle values might be an effect of solvent casting technique that exposed the OH hydrophilic groups of PLGA. Moreover, the enhanced water repellent nature of electrospun scaffolds can be attributed to their fibrous morphology: according to the Wenzel model for contact angles calculation, micro- and nano-roughness amplify the wettability performance of surfaces, meaning that already hydrophilic surfaces will become more wettable, while the already hydrophobic ones will become more water repellent [54]. This can be attributed to the increased surface area of electrospun scaffold compared to solvent-casted films. Hence, it is evident that architectural differences among electrospun and solvent-casted samples can lead to a variation in scaffolds’ wettability.

The comparison between the hydration profiles of electrospun scaffolds and their contact angles, suggest an absence of correlation between the contact angle values and the water uptake capacity of the whole scaffold. Contrary to contact angles, for electrospun scaffolds, PLGA content highly impacts the hydration rate and water uptake percentage at equilibrium, as shown by the hydration profiles in Figure 5 of ES4 (100% PLGA) and ES3 (PCL 85% + PLGA 15%), ES5 (PLGA 95% + PCL 5%), and ES6 (PLGA 85% + PCL 15%) scaffolds, due to the higher amount of hydroxylic water-bonding groups of PLGA. The lack of correlation between the results of water uptake and contact angle tests is explained considering that the latter is performed on a small scaffold area and is influenced by polymers’ properties and surface geometry, while water uptake relies on the interaction and response of the whole scaffold structure to water contact and is therefore highly influenced by the high surface-to-volume ratio of electrospun matrixes [54,55]. For example, ES4 scaffolds, made from 100% PLGA, with the smallest fibre diameter (0.607 ± 0.107 μm), and highest surface area, showed both a higher hydration rate and a higher percentage of water uptake. By simultaneously lowering the PLGA content and increasing the average fibre diameter, a decrease in water uptake was observed, as shown by the hydration profiles of ES3 (PCL 85% + PLGA 15%), ES5 (PLGA 95% + PCL 5%), ES6 (PLGA 85% + PCL 15%), ES2 (PCL 95% + PLGA 5%) and ES1 (100% PCL) in Figure 5. Solvent-casted samples do not benefit from a high surface area and their contact angles only slightly decreased by incorporating PLGA. Overall, a lower water uptake percentage in comparison with electrospun samples and a lack of significant differences between solvent-casted scaffold types were recorded.

The preliminary biologic evaluation of electrospun and solvent-casted scaffolds was carried out on episomal iPSCs. They are obtained from somatic cells reprogrammed into induced Pluripotent Stem Cells (iPSCs) using non-integrative episomal vector methods. This reprogramming process demonstrated a better safety profile compared with integrative methods using viruses [56]. As already mentioned, the scaffolds selected for the biologic evaluation were ES1 (PCL 100%), ES3 (PCL 85% + PLGA 15%), SC3 (PCL 85% + PLGA 15%) and SC5 (PLGA 95% + PCL 5%). This was done not to compare the two manufacturing techniques but to test samples that have pairwise similar morphology but differ in water uptake capacity (ES1 and ES3) and mechanical properties (SC3 and SC5). Moreover, the selected samples displayed decreasing contact angles and increasing YM, allowing for the evaluation of a wide range of physico-chemical and mechanical properties. ES1 and SC3 showed better cell viability compared to the other two scaffold types. Even though ES1 and ES3 showed comparable results on day 1, a rapid decline in cell viability was observed from day 3, while cells seeded on SC3 and SC5 showed from the beginning significant differences in metabolic activity. Electrospun scaffolds possess an ECM-like structure and are therefore able to guide and enhance cell attachment [57]. Thanks to this property, initial good cell viability was observed for electrospun scaffolds, but because of their hydrophobic properties (θ = 96.73 ± 1.30° for ES1 and θ = 106.83 ± 2.35 for ES3) cell proliferation decreased at later time points. On the other hand, solvent-casted samples are characterised by higher hydrophilicity, making them more attractive for cell attachment and proliferation. Nevertheless, SC3 showed the best performance among all samples, and this could be ascribed not only to its wettability but also to its YM. It was reported that softer cell culturing substrates are able to promote iPSC attachment and growth [58]. In fact, even if SC5 had comparable wettability to SC3 (θ = 76.60 ± 4.19° and 87.47 ± 0.55°, respectively) its stiffness was double (YM SC3 = 8.03 ± 2.57 MPa; YM SC5 = 20.23 ± 7.68 MPa), thereby explaining the lower cell viability compared to the control. Notably, ES1 displayed soft mechanical properties (YM = 3.25 ± 0.52 MPa), but its hydrophobicity seemed to play an important role as well. Hence, the combination of appropriate morphology, mechanical properties and wettability favoured iPSCs attachment and proliferation.

Quantitative PCR was performed to investigate the cellular response to the extracellular matrix (ECM) microenvironment mediated by integrin adhesion, which is of fundamental importance to obtain a preliminary indication of iPSCs maintenance of their pluripotency and/or possible differentiation. NANOG and OCT4 are both involved in the maintenance of pluripotency: NANOG has a major role in self-renewal, and OCT4 is one of Yamanaka’s factors and a main gene for pluripotency establishment and maintenance. All the analysed integrins have been reported as highly expressed in iPSCs [59], but while subunits α6, αV and β5 displayed a very similar expression pattern, ES1 scaffold showed a higher ITGβ1 expression (Figure 7b,c). Since ITGβ1 appears to be involved in the pluripotency maintenance of the inner cell mass, its high expression might be due to the presence of a pluripotency status. Concerning SC3 scaffold, on day 3 significant differences in the expression of NANOG were highlighted, resulting being more expressed in the scaffold seeded cells rather than the control (Figure 8b), while the integrins and OCT4 expression remained stable and the difference between the scaffold and control cells was not significant. Overall, the expression pattern of iPSCs seeded on electrospun and solvent-casted samples were similar; hence, both scaffold types did not affect iPSCs pluripotency and did not lead to spontaneous differentiation, making them suitable for the expansion of pluripotent stem cells. iPS cells can virtually differentiate in any somatic cell lineage, depending on the type of external stimulation. The scaffolds physico-chemical properties, with particular reference to YM, were compatible for cartilage, bone or blood vessel regeneration purposes. As reported in the literature, cartilage has low regenerative capacity and iPSCs are highly attractive for cartilage regeneration, even if in vitro differentiation of iPSCs into chondrocytes is highly complex because iPCs are extremely immature [60]. Even in the case of bone, which is a tissue generally capable of self-repair for example following a fracture, iPSCs represent a theoretically unlimited source of osteoblasts. It is reported in the literature that differentiation of pluripotent stem cells typically results in heterogeneous cellular populations, and even the presence of only a small fraction of osteoblasts can yield positive results in assays of osteoblast gene expression and mineralization [61]. Eventually, endothelial cells derived from human-induced pluripotent stem cells are reported in the literature with the purpose of large blood vessel regeneration [62].

## 5. Conclusions

Porous scaffolds made of PCL, PLGA, and their blends, manufactured through the two extremely different techniques of electrospinning and solvent casting, were characterised. As expected, the manufacturing technique affects scaffold architecture, mechanical properties and wettability. The physical–chemical characterization of electrospun scaffolds demonstrated that the blending of PCL and PLGA could lead to irregular fibre formation when PCL is the predominant polymer, whereas tailoring of the fibre diameter is possible by modulating PCL concentration in PLGA-based blends. Moreover, the use of PCL as a plasticising agent allowed for a significant reduction in the elastic modulus. In solvent casting/particulate leaching scaffolds, high PLGA content reduced surface microporosity, whereas intermediate PLGA content had a positive effect on contact angles that were lowered, thereby showing hydrophilic behaviour in PCL-based scaffolds. The mechanical properties are compatible not only for cardiovascular tissue engineering applications, but also with other soft tissues (e.g., cartilage, ligaments, mucosa), highlighting the versatility of the manufactured scaffolds. Concerning mechanical properties, PCL displayed a positive plasticising effect, but the inhomogeneity of the scaffolds was a main issue in precisely determining elastic moduli. From the biological standpoint, thanks to their higher wettability and appropriate Young moduli, solvent-casted samples seemed to be superior in ensuring cell attachment and proliferation, as long as excessive stiffness of the scaffold is avoided, whereas the ability to maintain iPSCs pluripotency status resulted similar for electrospun and solvent-casted samples. Nevertheless, further biological evaluation needs to be performed to determine if the selected scaffolds are a suitable support to induce the differentiation of iPSC in defined cell lines.

## Figures and Tables

**Figure 1 pharmaceutics-15-02736-f001:**
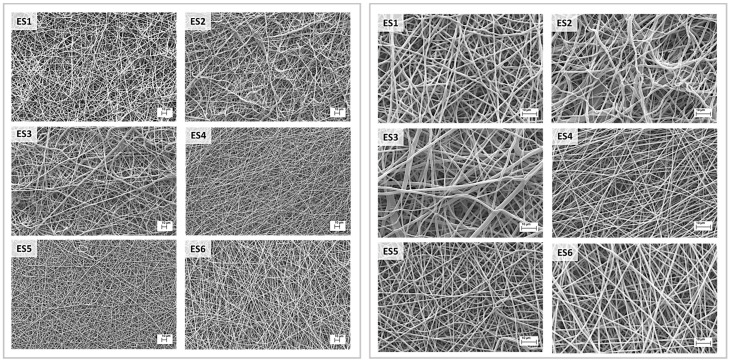
SEM images of the electrospun scaffolds made of different PCL/PLGA blends: (ES1) PCL100%; (ES2) PCL95% + PLGA5%; (ES3) PCL85% + PLGA15%; (ES4) PLGA100%; (ES5) PLGA95% + PCL5%; (ES6) PLGA85% + PCL15%. Left panel at 1000× magnifications (scale bar 10 µm) and right panel at 3000× magnifications (scale bar 10 µm).

**Figure 2 pharmaceutics-15-02736-f002:**
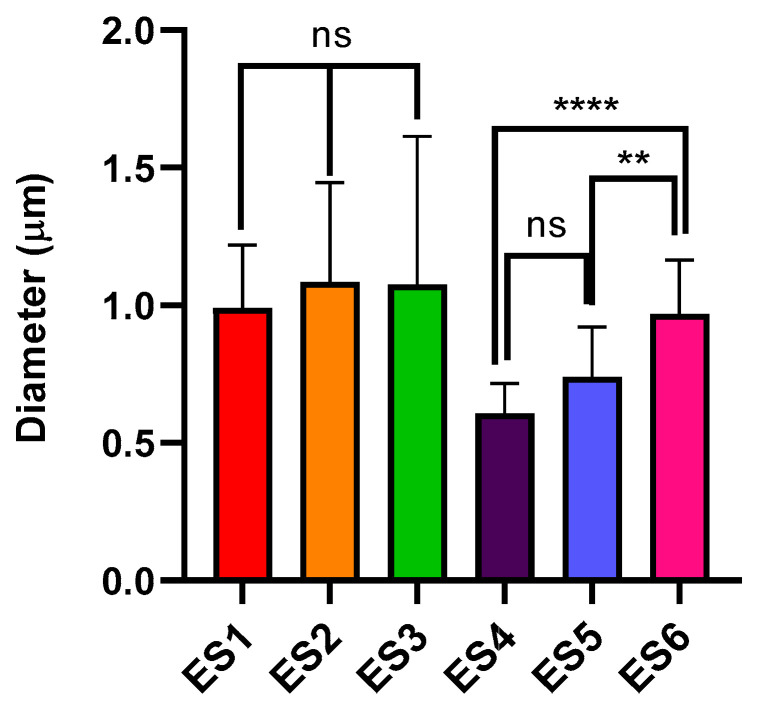
Plot depicting the average fibre diameter bars of scaled-up electrospun scaffolds. Data are presented as average ± standard deviation (n = 50). Statistical analysis was performed with one-way ANOVA with Tukey’s multiple comparison (ns = *p* > 0.05; ** = *p* < 0.01; **** = *p* < 0.0001).

**Figure 3 pharmaceutics-15-02736-f003:**
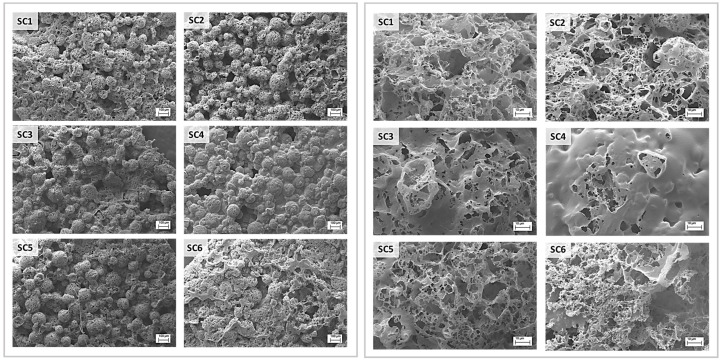
SEM images of the solvent-casted scaffolds made of different PCL/PLGA blends: (SC1) PCL100%; (SC2) PCL95% + PLGA5%; (SC3) PCL85% + PLGA15%; (SC4) PLGA100%; (SC5) PLGA95% + PCL5%; (SC6) PLGA85% + PCL15%. Left panel at 200× magnifications (scale bar 100 µm) and right panel at 3000× magnifications (scale bar 10 µm).

**Figure 4 pharmaceutics-15-02736-f004:**
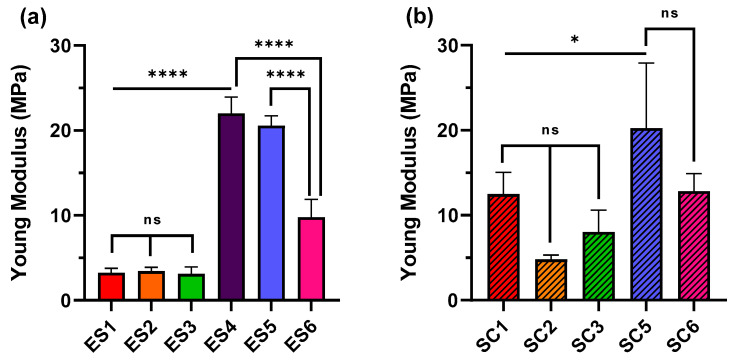
Comparison of Young moduli between (**a**) electrospun scaffolds and (**b**) solvent-casted scaffolds, with different PLGA and PCL polymers composition. Results are reported as average ± standard deviation. Statistical analysis was performed through one-way ANOVA analysis with Tukey’s multiple comparison (ns = *p* > 0.05, * = *p* < 0.05, **** = *p* < 0.0001).

**Figure 5 pharmaceutics-15-02736-f005:**
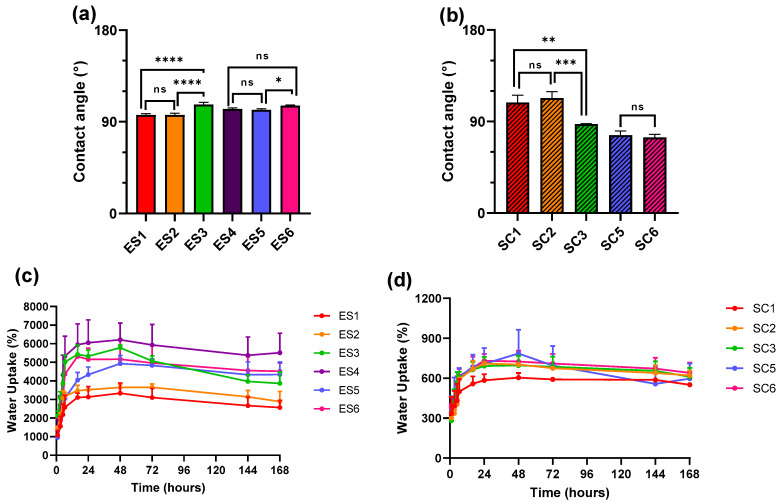
Statistical comparison of contact angle values between (**a**) electrospun scaffolds and (**b**) solvent-casted samples. Hydration profile of (**c**) electrospun scaffolds and (**d**) solvent-casted scaffolds in artificial interstitial fluid at pH = 7.4 and at temperature of 37 °C. Results are reported as average contact angle or water uptake% ± standard deviation (ns = *p* > 0.05, * = *p* < 0.05, ** = *p* < 0.01, *** = *p* < 0.001, **** = *p* < 0.0001).

**Figure 6 pharmaceutics-15-02736-f006:**
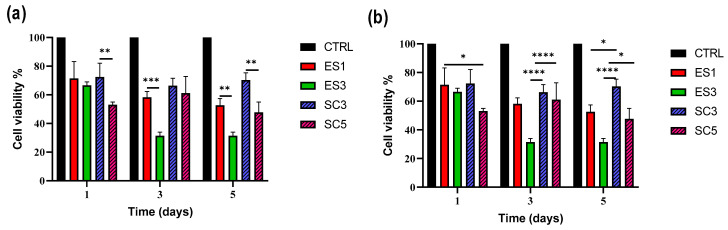
iPSC viability % after seeding on electrospun and solvent-casted scaffolds and incubating for 1, 3 and 5 days at 37 °C in normoxia conditions. Results are reported as average ± standard deviation. Statistical analysis was performed through one-way ANOVA analysis with Tukey’s multiple comparison (* = *p* < 0.05, ** = *p* < 0.01, *** = *p* < 0.001, **** = *p* < 0.001) and is displayed as follows: (**a**) statistical comparison between scaffolds produced with the same technique; (**b**) comparison between scaffolds produced using different techniques. If not specified, no significant difference was detected. Based on XTT assay, further RT-qPCR analysis was carried out on those scaffolds demonstrating higher cell viability for each manufacturing technique, i.e., ES1 and SC3.

**Figure 7 pharmaceutics-15-02736-f007:**
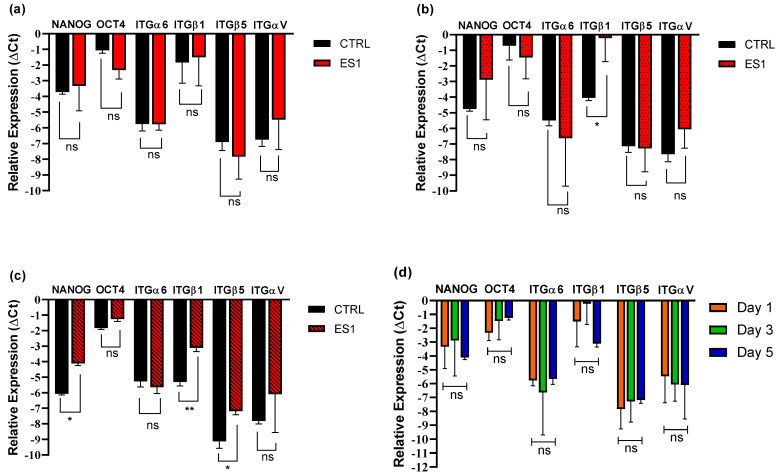
Plots depicting gene expression of (**a**–**c**) iPSC incubated on electrospun scaffolds ES1 (PCL 100%) as compared to control on day 1, 3, 5, respectively; (**d**) iPSC incubated on ES1 scaffolds as compared to day 1, 3 and 5. Statistical analysis using one–way ANOVA analysis with Sidak’s correction (significance of the results by the *p*–value as ns = *p* > 0.05, * = *p* < 0.05, ** = *p* < 0.01).

**Figure 8 pharmaceutics-15-02736-f008:**
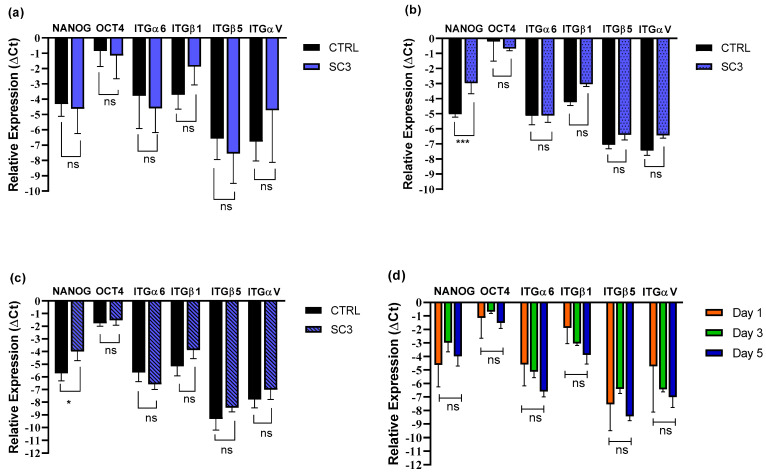
Plots depicting gene expression of: (**a**–**c**) iPSCs incubated on solvent-casted scaffolds SC3 (PCL 85% + PLGA 15%) as compared to control on day 1, 3, 5, respectively; (**d**) iPSCs incubated on SC3 scaffolds as compared to day 1, 3 and 5. Statistical analysis by one-way ANOVA analysis with Sidak’s correction (significance of the results by the *p*-value as ns = *p* > 0.05, * = *p* < 0.05, *** = *p* < 0.001).

**Table 1 pharmaceutics-15-02736-t001:** PCL and PLGA blends composition investigated in scaffold preparation via electrospinning (ES) and solvent casting/particulate leaching (SC).

Batch	Polymer Solution Composition	
	PCL (*w*/*w*)	PLGA (*w*/*w*)
ES1	100%	-
ES2	95%	5%
ES3	85%	15%
ES4	-	100%
ES5	5%	95%
ES6	15%	85%
SC1	100%	-
SC2	95%	5%
SC3	85%	15%
SC4	-	100%
SC5	5%	95%
SC6	15%	85%

**Table 2 pharmaceutics-15-02736-t002:** List of genes primers used in RTq-PCR.

Gene Name	Sequence Name	Primer Sequence
GAPDH	GAPDH RV	5′-ACCAGGAAATGAGCTTGACAAA-3′
GAPDH	GAPDH FW	5′-TCAAGAAGGTGGTGAAGCAGG-3′
NANOG	NANOG RV	5′-TTCCAGGTCTGGTTGCTCCACATT-3′
NANOG	NANOG FW	5′-TGGCCGAAGAATAGCAATGGTGTG-3′
OCT4	OCT4 RV	5′-GCCGCAGCTTACACATGTTCTTGA-3′
OCT4	OCT4 FW	5′-CGAGCAATTTGCCAAGCTCCTGAA-3′
ITGα6	Integrin Alpha 6 RV	5′-TTCCTGCTTCGTATTAACATGCT-3′
ITGα6	Integrin Alpha 6 FW	5′-ATGCACGCGGATCGAGTTT-3′
ITGβ1	Integrin Beta 1 RV	5′-TCCCCTGATCTTAATCGCAAAAC-3′
ITGβ1	Integrin Beta 1 FW	5′-GTAACCAACCGTAGCAAAGGA-3′
ITGβ5	Integrin Beta 5 RV	5′-TGGCGAACCTGTAGCTGGA-3′
ITGβ5	Integrin Beta 5 FW	5′-TCTCGGTGTGATCTGAGGG-3′
ITGαV	Integrin Alpha V RV	5′-TCTGCTCGCCAGTAAAATTGT-3′
ITGαV	Integrin Alpha V FW	5′-GCTGTCGGAGATTTCAATGGT-3′

**Table 3 pharmaceutics-15-02736-t003:** Results of ImageJ analyses on SEM images: fibre diameters, pore diameters, percentage of porosity electrospun scaffolds (*) and solvent-casted scaffolds (**) made of polymer blends are reported as average ± standard deviation for fibre diameter n = 50; for pore diameter n = 200). *** NA = Not Applicable.

Batch	Polymer Blend Composition (*w*/*w*)	Fibre Diameter (µm)	Pore Diameter (µm)	Porosity (%)
* ES1	PCL 100%	0.992 ± 0.228	3.15 ± 2.55	37.58
ES2	PCL 95% + PLGA 5%	1.085 ± 0.360	3.47 ± 3.32	40.44
ES3	PCL 85% + PLGA 15%	1.076 ± 0.537	3.15 ± 2.70	33.46
ES4	PLGA 100%	0.607 ± 0.107	3.62 ± 3.04	51.60
ES5	PLGA 95% + PCL 5%	0.740 ± 0.181	3.59 ± 2.74	48.25
ES6	PLGA 85% + PCL 15%	0.969 ± 0.195	3.51 ± 2.81	40.85
** SC1	PCL 100%	*** NA	3.10 ±11.10	61.42
SC2	PCL 95% + PLGA 5%	NA	3.13 ± 9.91	48.50
SC3	PCL 85% + PLGA 15%	NA	3.67 ± 8.87	41.92
SC4	PLGA 100%	NA	NA	NA
SC5	PLGA 95% + PCL 5%	NA	2.85 ± 7.94	32.64
SC6	PLGA 85% + PCL 15%	NA	2.88 ± 6.62	23.01

**Table 4 pharmaceutics-15-02736-t004:** Young moduli and contact angle values of scaled-up electrospun scaffolds (*) and solvent-casted scaffolds (**), reported as average ± standard deviation. *** NA = Not Applicable.

Batch	Polymer Blend Composition (*w*/*w*)	Young Modulus (MPa)	Contact Angle (°)
* ES1	PCL 100%	3.25 ± 0.52	96.73 ± 1.30
ES2	PCL 95% + PLGA 5%	3.43 ± 0.46	96.63 ± 1.72
ES3	PCL 85% + PLGA 15%	3.11 ± 0.83	106.83 ± 2.35
ES4	PLGA 100%	22.02 ± 1.93	102.47 ± 1.26
ES5	PLGA 95% + PCL 5%	20.56 ± 1.15	101.70 ± 0.98
ES6	PLGA 85% + PCL 15%	9.80 ± 2.09	105.80 ± 0.78
** SC1	PCL 100%	12.48 ± 2.56	108.60 ± 7.23
SC2	PCL 95% + PLGA 5%	4.81 ± 0.494	113.23 ± 6.23
SC3	PCL 85% + PLGA 15%	8.03 ± 2.57	87.47 ± 0.55
SC4	PLGA 100%	*** NA	NA
SC5	PLGA 95% + PCL 5%	20.23 ± 7.68	76.60 ± 4.19
SC6	PLGA 85% + PCL 15%	12.83 ± 2.08	74.47 ± 3.00

## Data Availability

Data are contained within the article and Appendix A.

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
