# Peer review of "Investigation on Electrospun and Solvent-Casted PCL-PLGA Blends Scaffolds Embedded with Induced Pluripotent Stem Cells for Tissue Engineering"

_pharmaceutics, 2023, doi:10.3390/pharmaceutics15122736_

Round 1

Reviewer 1 Report

Comments and Suggestions for Authors

The manuscript titled ‘Investigation on electrospun and solvent casted PCL-PLGA blends scaffolds embedded with induced pluripotent stem cells for tissue engineering’ is comprehensive and well organized. The manuscript might be accepted in the present form.

Comments on the Quality of English Language

NA

Author Response

Answer to reviwer 1 is uploaded here below.

Reviewer 2 Report

Comments and Suggestions for Authors

This work involves the materials processed with two very diverse techniques and two polymers, blended in different ratios. The authors performed a lot of experimental work to optimize the electrospinning and casting processing conditions. The work is presented as a comparison of many systematically tested materials.

Some comments are listed below:

1)    It is clear that electrospinning and solvent casting techniques allow obtaining porous materials, with different structural features (size, interconnection, etc). However, it is important to highlight the enormous technological differences between both processing techniques, in relation to their protocols, scaling-up potential, etc. The authors should incorporate a brief comparative evaluation in the Introduction (not as a Conclusion).

2)    What are the chemical/mechanical/degradability reasons for the choice of PCL-PLGA blends as a basis for cardiac tissue scaffolds? Besides the versatility of these polymers for scaffolds in general, their particular combination is proposed for cardiac tissue in this work. In this sense, the objective of the work is not clear. Why do the authors blend?

-One can imagine that the authors would like to modify hydrophobic PCL in order to improve the surface for cell interactions. However, the incorporation of PLGA increases the contact angle and worsen the cell viability (ES1 vs. ES3), and there are no (valid) hypothesis of why this is happening. Besides, ES1 contact angle seems unusual for typical PCL electrospun membranes.

- Another reason for blending could be to tune the degradability. However, this is not contrasted with bibliographic data for the proposed tissue. Please check and discuss other works, that recently reported electrospun scaffolds based on PCL 80.000 and PLGA 82:12 (e.g. 10.3390/ma14174773).

3) In Sections 3.5 and 3.6 (named alike) the authors compare ES1 and SC3, i.e. both the composition and the manufacturing technique are drastically modified.

On the one hand, only-PCL electrospun membrane (ES1) is a very common material, so that results can be maybe compared with available literature.

On the other hand, the structural target is also unclear. This fact was made explicit by the authors in the lines 638-642:  “A comparison among the scaffolds manufactured with the two techniques led to highlight how significantly the manufacturing techniques affect the scaffolds morphology, showing that polymer blend compositions should be chosen also according to the selected technique fabrication method.” I would add… According to the tissue to be replaced… do we need interconnected porosity and a fibrous structure, or a film with holes?

4)    It would be more appropriate to show fiber size distributions than average values. Maybe, fibers size comparisons, e.g. between ES2 (1.085 ± 0.360 μm) and ES3 (1.076 ± 0.537 μm) make more sense, to explain the differences in the physical properties measured. Otherwise, differences seem extremely slight.

5)    Given that electrospun membranes generate 3D interconnected porosity, pore diameter measurements can only be estimated approximately from a 2D image. The 0,01mm sensitivity of the measures reported in Table 3 does not reflect this fact. The authors should at least mention the existence of more appropriate techniques for the analysis of porosity in electrospun samples, and highlight that apparent porosity values based on relative areas of images are used, for comparative purposes. However, comparison would be only valuable between electrospun samples. It makes no sense to compare pore sizes that are structurally so different, such as those obtained by these techniques, at least if the comparison is based on imaging. 

6)    Electrospinning was performed during 30 minutes at 0,1 mL/h. Can it be considered a 3D scaffold as stated in line 144? Or could it be potentially scalable? What was the thickness? 

Author Response

Point by point answers to reviewer 2 are uploaded here below. 

Reviewer 3 Report

Comments and Suggestions for Authors

The main concern about this project is its novelty. Clearly describe the novelty of your project. Why it is important in the realm of tissue engineering? What insights it can give to the readers? 

Provide background information in the abstract and add some numerical data. The functional differentiation assessment of iPSCs is lacking. You need to repeat contact angle measurements for SC5 and SC6 samples. How do you justify the reduction in surface wettability in two hydrophobic polymers? You need to perform cell adhesion studies as well. Discussion, introduction, and conclusion sections need to be rewritten. They are too long. Write to the point.  

Comments on the Quality of English Language

The English content needs extensive revision. 

Author Response

Point by point answers to reviewer 3 are uploaded here below.

Reviewer 4 Report

Comments and Suggestions for Authors

This is a very interesting study overall, and the presentation of the findings is clear and well-structured. This manuscript can be very useful for researchers in the relevant research areas. Yet, a few minor issues have to be addressed to make this article suitable for publication:

1). Line 624, concerning the melting and glass transition temperature of PCL, did you measure it? If not kindly add the references. In my opinion, the reported values were higher than the actual values. Line 397, kindly remove the symbol "%." 

2). Line 590-593: Besides the mentioned parameters, nature of the high-voltage (AC or DC) also influences the electrospinning process. If AC power source is used, the waveform and frequency of AC high-voltage also impact the electrospinning process in terms of electrospun fiber morphology and productivity. Therfore it would be nice if the authors cite these articles (https://doi.org/10.1016/j.mtchem.2022.101025) (https://doi.org/10.1016/j.matdes.2021.110308).

Author Response

Point by point answers to reviewer 4 are uploaded here below

Reviewer 5 Report

Comments and Suggestions for Authors

The text from line 37 to 143 has to be shortened, as this is not a review paper and only relevant literature should be mentioned briefly.

From the introduction I thought that electrospinning and casting would be combined. Instead, both production methods are used separately. As described in the introduction, this is actually state of the art and nothing new, the combination would have been something new. Nor do I understand why these two techniques and the scaffolds they produce are being compared. The differences between the geometric and mechanic parameters are more or less clear and to be expected. Interestingly, the viability results show almost no differences (although the text strangely says otherwise). 

In any case, the method of determining porosity and pore size should be described in detail. Especially in the case of fibre-based scaffolds, there are many different approaches and they give different results, so the description of the method used is very important (line 230). In addition to pores, the surface/volume ratio is sometimes mentioned, as a high surface area can be good for cell seeding.

The discussion is very detailed, which is rarely the case. This is good. However, one should not only discuss one's own results, but also be open to other approaches. It might be worth discussing why this attempt to compare electrospinning and casting was made, why it is relevant and how it fits into the current literature.

All in all, a very well written paper, but it is not clear why this extensive research was done and what is to be learned from this comparison of electrospinning and casting.

Author Response

Answer to reviewer 5 is reported here below.

Round 2

Reviewer 2 Report

Comments and Suggestions for Authors

*The authors justified the compositional conceptual bases of the work, which were made explicit in the manuscript. 
* For nex times, it is advisable to mark (in yellow) the new references incorporated in the list of references of the new version. 
It is noticeable that the authors decide not to include the discussion and comparison of their results with those of strikingly similar systems: electrospun scaffolds based on PCL 80,000 and PLGA 82:12 intended for vascular scaffolds. (e.g. doi: 10.3390/ma14174773, and others)
* The authors included the Fibers diameters distributions in the SI. It is suggested to include units in the graphics (x axis). The graphs make it easy to visualize the differences between polymers, and how the effect of aggregates broadens the size distributions. 

Author Response

REVIEWER 2 Comments and Suggestions for Authors

REVIEWER COMMENT *The authors justified the compositional conceptual bases of the work, which were made explicit in the manuscript. 

ANSWER: no comment

REVIEWER COMMENT * For nex times, it is advisable to mark (in yellow) the new references incorporated in the list of references of the new version. 

ANSWER: the authors apologize for missing to yellow highlighting the new references in the list.

REVIEWER COMMENT It is noticeable that the authors decide not to include the discussion and comparison of their results with those of strikingly similar systems: electrospun scaffolds based on PCL 80,000 and PLGA 82:12 intended for vascular scaffolds. (e.g. doi: 10.3390/ma14174773, and others)

ANSWER: The authors thank the reviewer for her/his punctual comment and they wish to explain in details why they chose not to include the discussion and comparison of their results with those electrospun scaffolds reported in the paper doi: 10.3390/ma14174773. The process conditions and parameters used by the authors were different from those applied by Bazgir M. As a consequence, the characteristics of the elecrospun matrices manufactured by Bazgir and coworkers couldn’t be similar or comparable to those of electrospun matrices manufactured by Rosalia M. and coworkers.

Briefly:

Bazgir and coworkers:  The polymeric solutions were prepared by dissolving 4.5 g of PCL pellets in 25.5 g of chloroform and 3 g of PLGA in 13.5 g of THF and 13.5 g of DMF (50:50). The solutions were placed on a magnetic stirrer in a sealed glass container for a minimum of 16 h; next, when the polymer pellets were entirely dissolved in the solution, the glass vials were placed in the ultrasonic bath for an additional 2 h to eliminate any bubbles that had been produced during the mixing procedure.

Rosalia M. and coworkers: “Polymer solutions to be electrospun were PCL 14% w/v in DCM/DMF 75:25, PLGA 10% w/v in DCM/DMF 75:25.”

Bazgir and coworkers dissolved PCL in chloroform at 17.65 w/w % concentration, while Rosalia and coworkers used methylen chloride as PCL solvent and the final polymer solution concentration was 14 % w/v.

Chloroform boiling point is 61.2° C while methylene chloride boiling point is 39.6° C. This makes a significant difference in electrospinning process because fiber drying rate depends also on polymer solvent boiling point and it affects the fiber diameter and porosity.  Moreover, Bazgir M. and coworkers added THF to DMF as PLGA solvent, while Rosalia M.  and coworkers did not use THF as PLGA solvent.

Bazgir M. and coworkers used a 20 Gauge needle to elctrospin the polymer solution, while Rosalia M. and coworkers used a 25 Gauge needle. This makes a difference as for the amount of polymer solution ejected in the unit time and for the fiber diameter and porosity. I.e. average fiber diameter of Bazgir and coworkers paper was in the rank 1.5 – 2.5 micron, while fiber diameter of Rosalia and coworker paper was between 0.60 and 0.96 micron. Concerning pore size and surface porosity, Bazgir and coworkers reported the changes in surface pore size and the changes in surface porosity percentages of their electrospun scaffolds during in vitro degradation test, while Rosalia and coworkers reported the pore size and porosity % values of their electrospun scaffolds. Thus, it was not possible to compare these parameters.

For these reasons the authors think the electrospun scaffolds of these two papers are neither similar, nor comparable.

However, a statement together with the reference was added in the current revised version of manuscript discussion, to explain how same electrospinning technique can lead to achieve different structures.

REVIEWER COMMENT: * The authors included the Fibers diameters distributions in the SI. It is suggested to include units in the graphics (x axis). The graphs make it easy to visualize the differences between polymers, and how the effect of aggregates broadens the size distributions. 

ANSWER: Following the reviewer comment, the authors transposed bin center (x axis) into micron in the current revised version of the manuscript.

Reviewer 3 Report

Comments and Suggestions for Authors

No comments. 

Comments on the Quality of English Language

Please improve the English content of this manuscript. For instance " The higher CEC value of PLGA is consistent with the polymer’s higher molecular weight, resulting in higher viscosity values if compared with same concentrated PCL polymer solutions" can be written like " The elevated CEC of PLGA aligns with the polymer's increased molecular weight, leading to higher viscosity values when compared to polymer solutions with the same concentration of PCL".

Author Response

REVIEWER 3 Comments and Suggestions for Authors

REVIEWER COMMENT: No comments. 

REVIEWER COMMENT: Comments on the Quality of English Language

Please improve the English content of this manuscript. For instance, "The higher CEC value of PLGA is consistent with the polymer’s higher molecular weight, resulting in higher viscosity values if compared with same concentrated PCL polymer solutions" can be written like "The elevated CEC of PLGA aligns with the polymer's increased molecular weight, leading to higher viscosity values when compared to polymer solutions with the same concentration of PCL".

ANSWER: English language was thoroughly revised along the manuscript.

Reviewer 5 Report

Comments and Suggestions for Authors

The introduction has been shortened to make the document a little shorter and clearer. At least some methods for pore characterization have been added. However, I would still like to see surface-to-volume data.

All in all, the same question remains as to why the two techniques of casting and electrospinning are being compared at all. This can be seen, for example, in Figure 6, where I do not see a systematic approach. Two different techniques are compared in terms of cell viability - of course you will get different results. It is not enough to say that a significant difference has been found, because that is what you would expect. You could do that with any production method. That this approach does not make sense can be seen from the graph in Figure 6 itself: The data point of sample ES1 is between SC3 and SC5 after 5 days. Is this a significant difference or how should it be interpreted? I think that at least these results should be presented in a different way than in the graph or with more data.

Author Response

REVIEWER 5: Comments and Suggestions for Authors

REVIEWER COMMENTS: The introduction has been shortened to make the document a little shorter and clearer. At least some methods for pore characterization have been added. However, I would still like to see surface-to-volume data.

All in all, the same question remains as to why the two techniques of casting and electrospinning are being compared at all. This can be seen, for example, in Figure 6, where I do not see a systematic approach. Two different techniques are compared in terms of cell viability - of course you will get different results. It is not enough to say that a significant difference has been found, because that is what you would expect. You could do that with any production method. That this approach does not make sense can be seen from the graph in Figure 6 itself: The data point of sample ES1 is between SC3 and SC5 after 5 days. Is this a significant difference or how should it be interpreted? I think that at least these results should be presented in a different way than in the graph or with more data.

ANSWER: Here below the authors respond point by point to the reviewer's detailed and precise comments

As far as the reviewer request to add surface-to volume data, the authors are aware these data can be obtained by adsorption methods based on adsorption of gas by solid surfaces, and they can further support the scaffold physical characterization. However, this analysis was not performed on the samples because we had not availability for this kind of instrumentation. Surface pores diameter and apparent percentage porosity are reported and compared in the paper as indicators of the scaffold porosity.

The reviewer asks why the two techniques of casting and electrospinning are being compared at all. The authors would like to clarify that these are two popular manufacturing techniques for TE scaffolds, and this is the reason the authors focused on these two manufacturing techniques. However, more than on manufacturing techniques, the focus of the work was on the material blends and their effect on the scaffold physical chemical characteristics and their correlation with biologic properties that were preliminarily evaluated. As explained by the authors in the paper introduction: Specifically, this preliminary study has the purpose to characterize these scaffolds from a physical, chemical, and biological point of view and to correlate their composition and characteristics to their biological properties”. Anyway, the authors are aware that comparing two different techniques in terms of cell viability of course you will get different results. The point is the correlation of the different results to the physical-chemical properties of the scaffolds. Furthermore, as highlighted from the results of biological tests, the ability to maintain iPSCs pluripotency status resulted similar for electrospun and solvent casted samples.

Concerning the reviewer comments addressed to the biologic evaluation of scaffolds, the authors would like to clarify that the biologic evaluation was a preliminary evaluation and was addressed as to support the physico-chemical characterization. Especially referring to biologic evaluation, the main goal was to test samples that have pairwise similar morphology but differ in water uptake capacity (ES1 and ES3) and mechanical properties (SC3 and SC5). Moreover, the selected samples displayed decreasing contact angles and increasing YM, allowing the evaluation of a wide range of physico-chemical and mechanical properties.

As for Figure 6 is concerned, the authors clarify that the Figure shows the results in term of iPSC viability % after seeding on electrospun and solvent casted scaffolds and incubating for 1, 3 and 5 days at 37°C in normoxia conditions. The systematic approach followed by the authors was to report all the data of cell viability referred to the selected scaffolds, and to split the results of statistical analysis in two parts, Figure 6a and Figure 6b as follows:  (a) statistical comparison between scaffolds produced with the same technique; (b) comparison between scaffolds produced using different techniques. The authors hope that these explanations clarify the concerns highlighted by the reviewer regarding Figure 6.

Moreover, the data in Figure 6 are thoroughly discussed in detail as follows: “….ES1 and SC3 showed better cell viability compared to the other two scaffold types: even though ES1 and ES3 showed comparable results at day 1, a rapid decline of cell viability is observed from day 3, while cells seeded on SC3 and SC5 showed from the beginning significant differences in metabolic activity. Electrospun scaffolds possess an ECM-like structure and are therefore able to guide and enhance cell attachment [56]. Thanks to this property, initial good cell viability was observed for electrospun scaffolds, but because of their hydrophobic properties (θ = 96.73∓1.30° for ES1 and θ = 106.83 ∓ 2.35 for ES3) cell proliferation decreased on later time points. On the other hand, solvent casted samples are characterized by higher hydrophilicity, being more attractive for cell attachment and proliferation. Nevertheless, SC3 showed the best performance among all samples, and this could be ascribed not only to its wettability but also to its YM. It was reported that softer cell culturing substrates are able to promote iPSC attachment and growth [57]. In fact, even if SC5 had comparable wettability to SC3 (θ = 76.60 ∓ 4.19° and 87.47 ∓ 0.55°, respectively), its stiffness was double (YM SC3 = 8.03 ∓ 2.57 MPa; YM SC5 = 20.23∓ 7.68 MPa) explaining the lower cell viability compared to the control. Noteworthy, ES1 displayed soft mechanical properties (YM = 3.25 ∓ 0.52 MPa), but its hydrophobicity seemed to play an important role as well. Hence, the combination of appropriate morphology, mechanical properties and wettability favoured iPSCs attachment and proliferation.”

Round 3

Reviewer 5 Report

Comments and Suggestions for Authors

Despite the authors' explanation, I would like to point out once again that I do not understand the purpose of the comparison and I do not believe that the comparison makes sense. Even if it did make sense, I don't understand why these two methods are being compared. They are popular, whatever popular means, but so are other methods, and there are many known methods for making scaffolds. I would also like to point out once again that I don't think the work itself is bad, but actually the application of one of the methods would be enough and it wouldn't be such a strangely artificial comparison. Normally it is enough to use exactly one method and compare the results or a parameter of the results with another method from another paper. When using this one method, you can also correlate the influence of input parameters with the results or determine any significant differences, but with two completely different methods?